# Neural parameter calibration and uncertainty quantification for epidemic forecasting

**Thomas Gaskin**[1,2]*, **Tim Conrad**[3], **Grigorios A. Pavliotis**[2], **Christof Schütte**[3,4]

**1** Department of Applied Mathematics and Theoretical Physics, University of Cambridge, Cambridge, United Kingdom, **2** Department of Mathematics, Imperial College London, London, United Kingdom, **3** Zuse Institute Berlin, Berlin, Germany, **4** Department of Mathematics and Computer Science, Freie Universität Berlin, Berlin, Germany

* trg34@cam.ac.uk

**Data Availability Statement:** Versioned under Github: https://github.com/ThGaskin/NeuralABM.

**Funding:** TG was funded by the University of Cambridge School of Physical Sciences VC Award via DAMTP and the Department of Engineering,

## Abstract

The recent COVID-19 pandemic has thrown the importance of accurately forecasting contagion dynamics and learning infection parameters into sharp focus. At the same time, effective policy-making requires knowledge of the uncertainty on such predictions, in order, for instance, to be able to ready hospitals and intensive care units for a worst-case scenario without needlessly wasting resources. In this work, we apply a novel and powerful computational method to the problem of learning probability densities on contagion parameters and providing uncertainty quantification for pandemic projections. Using a neural network, we calibrate an ODE model to data of the spread of COVID-19 in Berlin in 2020, achieving both a significantly more accurate calibration and prediction than Markov-Chain Monte Carlo (MCMC)-based sampling schemes. The uncertainties on our predictions provide meaningful confidence intervals e.g. on infection figures and hospitalisation rates, while training and running the neural scheme takes minutes where MCMC takes hours. We show convergence of our method to the true posterior on a simplified SIR model of epidemics, and also demonstrate our method's learning capabilities on a reduced dataset, where a complex model is learned from a small number of compartments for which data is available.

## Introduction

The COVID-19 pandemic has underscored the need for comprehensive epidemiological models. In a crisis, an effective government response is predicated on such models (1) being formulated using interpretable infection parameters, (2) being capable of accurately and quickly forecasting the dynamics of contagion, and (3) meaningfully capturing the uncertainty inherent in their projections [1–3]. It is this last point in particular—the inclusion of uncertainty quantification—that enables informed and transparent cost-benefit analyses, since it lets policy-makers assess the probable efficacy of different intervention strategies.

Ordinary differential equations (ODEs) play a key role in the study of mathematical epidemiology, since they often serve as the foundation for so-called *compartmental models*. Compartmental models capture the dynamics of patients progressing through the various stages of a

and supported by EPSRC grants EP/P020720/2 and EP/R018413/2. TC and CS were funded by the Deutsche Forschungsgemeinschaft (DFG) under Germany's Excellence Strategy through grant EXC-2046 The Berlin Mathematics Research Center MATH+ (project no. 390685689) and via the grant MODUS-COVID by the German Ministry for Education and Research (BMBF) (grant number 031L0302C). GP is partially supported by the Frontier Research Advanced Investigator Grant ERC grant Machine-aided general framework for fluctuating dynamic density functional theory. The funders had no role in study design, data collection and analysis, decision to publish, or preparation of the manuscript. There was no additional external funding received for this study.

**Competing interests:** The authors have declared that no competing interests exist.

disease—from, e.g., a susceptible and exposed state all the way to being infected, recovering, becoming immunised, or falling critically ill. The transition rates between these compartments model the biological and behavioral drivers of disease transmission and recovery. Based on Bernoulli's seminal work in the 1760s [4], the first compartmental models were introduced by Kermack & McKendrick [5]. Since then, compartmental ODE models have played a crucial role in offering a profound understanding of the transmission dynamics of infectious illnesses. However, ODE models are inherently deterministic and thus fail to account for stochastic characteristics of real-world transmission processes. Over the years they have therefore been evolved into stochastic variants, based e.g. on stochastic differential equations (SDEs) [6, 7], and, more recently, agent-based micro-models (ABMs), incorporating demographic data with full spatial resolution [8, 9]. Such models have been vital in assisting public health authorities in predicting outbreaks and implementing efficient disease control strategies in a variety of instances, including influenza, West Nile virus, childhood illnesses, SARS-CoV-1, rabies, sexually transmitted infections such as AIDS, and—most recently—COVID-19 [10–13]. In this work, we employ a computational method that combines the traditional ODE approach with a recently developed neural parameter estimation scheme [14] to recover the model parameters driving the dynamics of a disease from observable data. Neural parameter calibration uses neural networks to optimally calibrate a given model to available data, thereby capturing both the deterministic and stochastic components of contagion within a single modelling framework. Here, we adapt this method to a high-dimensional compartmental model of infectious diseases.

The existing body of research covers a variety of methods for data-based parametrisation of ODE models quantifying associated uncertainties, including hierarchical models, non-parametric techniques, ensemble techniques, and Bayesian approaches [15–18]. The ubiquitous Bayesian paradigm again unfolds into a rich tapestry of techniques, including Markov-Chain Monte Carlo methods (MCMC) [19] such as Hamiltonian Monte Carlo [20], or samplers based on Langevin dynamics, such as the Metropolis-adjusted Langevin algorithm (MALA) and its preconditioned variants [21–24], to name just a few. In the context of Covid-19, ensemble methods and approximate Bayesian computation have been used to model not only the viral dynamics, but also gauge the uncertainty arising from model misspecification, parameter uncertainty, or stochasticity by running multiple model instances and calculating statistics over the ensemble [25–27]. The commonality of all Bayesian parameter estimation and uncertainty quantification schemes is that they require sampling from a posterior distribution. However, the sampling paradigm has several major disadvantages: first, since the likelihood of a parameter is represented by its sampling frequency, high-dimensional inference can quickly become a costly affair. This is particularly true when the likelihood of a sample must be computed by solving the underlying ODE. Random-walk behaviour may render sampling in high dimensions computationally infeasible, and it is for this reason that modern MCMC samplers are often strongly gradient-driven. Second, the complex geometry of high-dimensional distributions may lead to samplers getting caught in local minima, and chains not reaching stationarity. This at times can be remedied by incorporating the topology of the distribution into a preconditioner, using e.g. the Hessian or Fisher information matrices [22]. However, calculating the Riemannian metric of a high-dimensional space can again become computationally expensive, especially if such information is only defined locally and thus needs to be recalculated at each point. Third, almost all sampling strategies require some sample rejection and burn-in periods, during which samples are discarded to ensure convergence of the Markov chains. Here again, whenever likelihoods must be obtained through expensive simulations, having to discard samples means wasting computational resources.

The strategy proposed in this paper falls within the Bayesian paradigm and yet circumvents these problems. It thus represents a notable improvement over existing techniques, both in

terms of computational efficiency and accuracy, and has previously been applied to a diverse set of problems, from estimating low-dimensional ODE parameters of economics models to entire network adjacency matrices in power grid dynamics and optimal transport [14, 28]. Being purely gradient-driven, no samples are rejected, a burn-in period is unnecessary, and the method quickly finds the modes of the distribution. The neural network parametrises the parameter space without the need for calculating Riemannian metrics. The loss function contains knowledge of the model equations, and likelihoods are estimated using simulation. Crucially, by performing multiple (parallelisable) training iterations, our methodology not only improves the accuracy of individual parameter estimations, but also offers a comprehensive representation of the likelihood distribution over the parameter space and thereby an understanding of the inherent uncertainty—a vital aspect in the formulation of policies in the face of intrinsic unpredictability, as is usually the case in a rapidly developing pandemic scenario.

The main purpose of this article is to demonstrate that this method, first presented in previous work, is straightforwardly applicable to applied epidemiological modelling of real-world data, while going beyond merely enhancing parameter accuracy; instead, we systematically address the frequently overlooked aspect of uncertainty in disease forecasting. We begin by presenting the mathematical foundations of our method on a simple model of epidemics, before revisiting a sophisticated compartmental model and recalibrating it to observation data of the spread of COVID-19 in Berlin.

## Methodology

In this section, we delve into the specifics of our methodology, starting with the underlying ideas and using a simple epidemic model as an example. The basic concepts have previously been described in [14], but will be summarised again in the following with a view to applying it to epidemic modelling.

Consider an Itô stochastic differential model with $N$ compartments of the dynamics of some contagious disease,

$$\frac{\mathrm{d}\mathbf{y}}{\mathrm{d}t} = f(\mathbf{y}(t); \mathbf{\Lambda}) + \sigma(\mathbf{y}(t); \mathbf{\Lambda})\boldsymbol{\xi}_t. \tag{1}$$

Here, $\mathbf{y}(t) \in \mathbb{R}^N$ is the $N$-dimensional state vector describing the model at time $t$, $f$ and $\sigma$ are the drift vector and diffusion matrix, respectively, $\mathbf{\Lambda} \in \mathbb{R}^p$ a vector of scalar parameters, and $\boldsymbol{\xi}_t$ an $N$-dimensional white noise process. Spatial dependence of $\mathbf{y}$, leading to (stochastic) partial differential equations, is not considered in this work but has been elsewhere [28]. We note that our method is not dependent on the choice of the stochastic integral used in the model equations.

Given a time series $\mathbf{T}$ comprising $L$ observations of $\mathbf{y}$, $\mathbf{T} = (\mathbf{y}_1, \ldots, \mathbf{y}_L)$, our goal is to infer the parameters $\mathbf{\Lambda}$. To this end we train a neural network $u_\theta : \mathbb{R}^{N \times B} \to \mathbb{R}^p$, where the batch size $B \geq 1$ represents the number of time series steps that are passed as input and $\boldsymbol{\theta}$ the neural network parameters, to produce a parameter estimate $\hat{\mathbf{\Lambda}} = (\hat{\lambda}_1, \ldots, \hat{\lambda}_p)$ that, when inserted into the model equations (1), reproduces the observations $\mathbf{T}$. The neural network is trained using a loss function (such as a weighted least squares residual)

$$J\left(\hat{\mathbf{\Lambda}} \mid \mathbf{T}\right) = J\left(\hat{\mathbf{T}}(\hat{\mathbf{\Lambda}}) \mid \mathbf{T}\right), \tag{2}$$

where $\hat{\mathbf{T}}(\hat{\mathbf{\Lambda}})$ is the time series obtained by integrating Eq (1) using the estimated parameters.

The likelihood of any estimate is then simply proportional to

$$\rho\left(\hat{\mathbf{\Lambda}} \mid \mathbf{T}\right) \propto e^{-J}. \tag{3}$$

As $\hat{\mathbf{\Lambda}} = \hat{\mathbf{\Lambda}}(\boldsymbol{\theta})$, we may calculate the gradient $\nabla_{\boldsymbol{\theta}} J$ and use it to optimise the internal parameters of the neural net using a backpropagation method of choice. Calculating $\nabla_{\boldsymbol{\theta}} J$ thus requires differentiating the predicted time series $\hat{\mathbf{T}}$, and thereby the system equations, with respect to $\hat{\mathbf{\Lambda}}$. In other words: the loss function contains knowledge of the dynamics of the model. Finally, the true data is once again input to the neural net to produce a new parameter estimate $\hat{\mathbf{\Lambda}}$, and the cycle starts afresh. Note that the gradient descent is not applied to the parameter space directly, but to its reparametrisation as a neural network. The optimal reparametrisation can be obtained through optimising the hyperparameters of the neural network architecture, that is, the depth, size, and structure of the layers, the use of biases, and the choice of activation functions.

As the net trains, it traverses the parameter space, calculating a loss at each point. Unlike in MCMC, the posterior density in our approach is not constructed by considering the frequency with which each point is sampled, but rather calculated directly via the loss function at that point (cf. [14]). This entirely eliminates the need for rejection sampling or a burn-in time: at each point, the true value of the likelihood is obtained, and sampling a single point multiple times gives no additional information, leading to a significant improvement in computational speed. Since the stochastic sampling process is entirely driven by the gradient of $J$, the regions of high probability are typically found much more rapidly than with a random sampler, leading to a high sample density around the modes of the target distribution.

We thus track the neural network's path through the parameter space and gather the loss values it calculates along the way. Multiple training runs can be performed in parallel, and each chain is terminated once it reaches a stable minimum. The likelihood for each parameter is given by

$$\rho(\hat{\lambda}_i \mid \mathbf{T}) = \int p\left(\hat{\mathbf{\Lambda}} \mid \mathbf{T}\right) d\hat{\mathbf{\Lambda}}_{-i}, \tag{4}$$

where the $-i$ subscript indicates omission of the $i$-th component in $\hat{\mathbf{\Lambda}}$ in the integration. In high dimensions, calculating the joint distribution can become computationally infeasible, and we can approximate the likelihood function by calculating the two-dimensional joint density of the parameter estimate and the likelihood, $p(\lambda_i, e^{-J})$ and then integrating over the likelihood,

$$\rho(\hat{\lambda}_i \mid \mathbf{T}) \approx \int p\left(\hat{\lambda}_i, e^{-J}\right) d(e^{-J}). \tag{5}$$

By Bayes' rule, the posterior marginal is then

$$p(\hat{\lambda}_i \mid \mathbf{T}) = \rho(\hat{\lambda}_i \mid \mathbf{T}) \times \pi^0(\hat{\lambda}_i)$$

with $\pi^0$ the prior density [29]. The only prior information available about the values of the parameters is that they are positive, hence in the following we will always assume uniform priors on $\mathbb{R}_+$. Running multiple chains in parallel increases the sampling density on the domain, ensuring convergence to the posterior distribution in the limit of infinitely many chains, independently of the choice of the prior.

## Illustration of a simple epidemic model

We first consider a synthetic example of a simple model of epidemics with three compartments, before turning to the main goal of this paper, which is to present an extensive analysis on a dataset of the spread of COVID-19 in Berlin. Given observations of susceptible (S), infected (I), and recovered (R) agents, assume the dynamics of the epidemic are given by

$$dS = -\beta SI dt - \sigma I dW_S$$
$$dI = (\beta S - \tau^{-1}) I dt + \sigma I dW_I \qquad (6)$$
$$dR = \tau^{-1} I dt,$$

where $(\beta, \tau, \sigma) \in \mathbb{R}^3_+$ are the infection, recovery, and noise parameters respectively, and $W_S$, $W_I$ are independent Wiener processes. We generate noisy observations of the time series $\mathbf{T}(t)$ = (S($t$), I($t$), R($t$)) using the parameters $\beta = 0.2$, $\tau = 14$, and $\sigma = 0.1$ (cf. Fig 1d), and try to recover the marginal densities on $\beta$ and $\tau$ given the observations, that is $\rho(\beta|\mathbf{T})$ and $\rho(\tau|\mathbf{T})$. The ground truth marginals we obtain by running a grid search on $(\beta, \tau)$ and calculating the likelihood at each grid point. We train the neural net using the loss function

$$J = \|\hat{\mathbf{T}} - \mathbf{T}\|_2^2 = -\log \rho(\mathbf{\Lambda} \mid \mathbf{T}) \qquad (7)$$

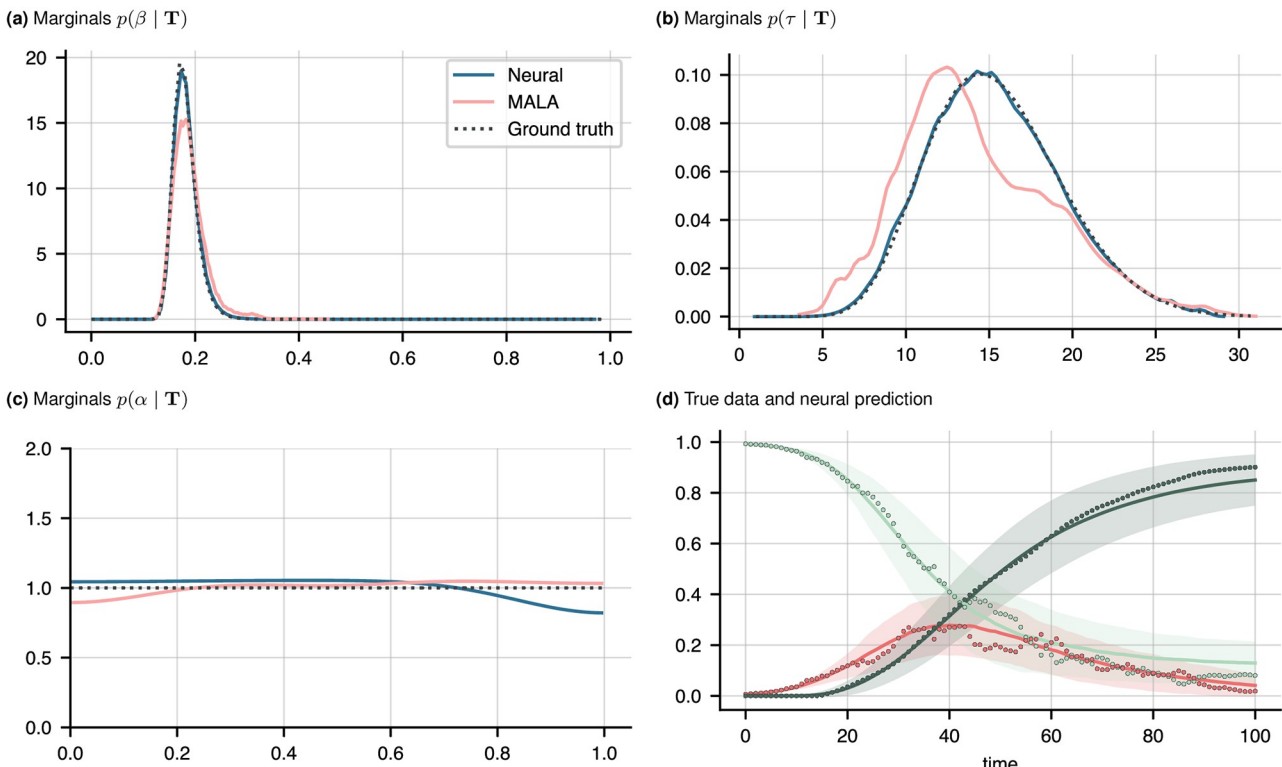

**Fig 1.** **(a)**–**(c)** Marginal densities on $(\beta, \tau, \alpha)$ for noisy SIR data, obtained from the neural scheme (blue) and the MALA sampler (pink). The ground truth (dotted line) was calculated using a simple grid search on $(\beta, \tau) \in [0, 1] \times [1, 30]$ with 10.000 grid points. **(d)** Predicted average time series $\langle \hat{\mathbf{T}} \rangle$ for the S (lightgreen), I (red), and R compartments. Shown are the predictions (solid line) and standard deviation (shaded area) generated by drawing 10.000 samples from the predicted joint distribution of $(\beta, \tau)$ using the neural scheme and solving the noiseless ODE model Eq (6) each. Also shown are the true data for each compartment (dots). The neural network was trained for 100 epochs from 300 different initial conditions, and 50 MALA chains were run until stationarity was reached, with a thinning factor of 5. Here, stationarity is defined via a Gelman-Rubin statistic of below 1.2, see S2 Fig in the S1 Appendix. Both the neural and MCMC samplers are parallelised.

and compare the predicted marginals to those generated by a preconditioned Metropolis-adjusted Langevin scheme (MALA) [22, 23]. The preconditioned MALA draws samples $\hat{\mathbf{\Lambda}}^i$ from the distribution

$$
\begin{aligned}
\hat{\mathbf{\Lambda}}^{i+1} \sim \frac{\epsilon^i}{2} \Bigg[ & G(\hat{\mathbf{\Lambda}}^i) \Big( \nabla \log \rho \Big( \hat{\mathbf{\Lambda}}^i \mid \mathbf{T} \Big) \\
& + \frac{L}{B} \sum_{k=1}^{L} \nabla \log \rho \Big( \mathbf{T}_k \mid \hat{\mathbf{\Lambda}}^i \Big) \Big) + \Gamma(\hat{\mathbf{\Lambda}}^i) \Bigg] \\
& + G^{\frac{1}{2}}(\hat{\mathbf{\Lambda}}^i) \mathcal{N}(0, \epsilon^i \mathbf{I}).
\end{aligned}
\tag{8}
$$

Here, $\epsilon^i$ is the step size at iteration $i$, $G(\mathbf{\Lambda})$ a preconditioning matrix as given in [23], and $\Gamma(\mathbf{x}) = \sum_j \frac{\partial G_{ij}}{\partial x_i}$. The series $\{\epsilon^i\}$ is chosen to be a decaying series with coefficient $\alpha < 1$. The algorithm is given in [23], which uses an approximate Fisher information matrix for the preconditioner, thereby maintaining good computational performance.

A hyperparameter sweep showed a two-layer neural network with 20 nodes per layer to provide optimal results. We use hyperbolic tangent activation functions on all except the last layer, where we use the modulus $|\cdot|$. The net is optimised using Adam [30] and a learning rate of 0.002. Best results were obtained using a batch size of $B = L$ (batch gradient descent). Results are shown in Fig 1a and 1b. Both the neural and MCMC approaches are initialised from multiple different values with sampling chains run in parallel. The initial values for $\beta$ are drawn from a uniform distribution on [0, 1], and those for $\tau$ from uniform distribution on [1, 30] (see S1 Appendix). The MCMC scheme is run with a burn-in time of 500 steps per chain, and a thinning factor of 5, meaning only every fifth sample is retained. We see that both the neural and Langevin schemes find the posterior marginals, though the neural estimates are more accurate.

To additionally test the neural method's ability to perform sensitivity analyses, we modify Eq (6) by adding a small perturbation to each term,

$$
\mathrm{d}\tilde{S} = \mathrm{d}S + \frac{\mathrm{d}t}{1000 + \alpha}
$$

(and similarly I and R), where $\alpha \in [0, 1]$, meaning that $\alpha$ is essentially irrelevant to the dynamics, and its marginal posterior approximately uniform on [0, 1]. As shown in Fig 1c, both schemes obtain the expected result.

We more formally compare the distribution accuracy in terms of the Hellinger distance to the ground truth,

$$
d_{\mathrm{H}}(\hat{p}, p) = \frac{1}{2} \int \left( \sqrt{\hat{p}(x)} - \sqrt{p(x)} \right)^2 \mathrm{d}x,
\tag{9}
$$

where $\hat{p}$ is the estimated density and $p$ the ground truth. We find the neural marginals for $\beta$ and $\tau$ to be two orders of magnitude closer to the ground truth than the MCMC estimates, see Table 1. At the same time, the neural scheme runs about an order of magnitude faster than the MCMC sampler, a fact that was previously observed in [14, 28]. As mentioned in the introduction, the MCMC scheme is slowed down by (1) a burn-in period, (2) redundant samples being discarded in the Metropolis step, and (3) the long sampling time required to reach stationarity. Each of these drawbacks the neural scheme manages to avoid.

**Table 1. Hellinger distances (Eq (9)) between the estimated and true marginals.**

|  | Neural | MALA |
|---|---|---|
| Hellinger distance $\beta$ | 5e–4 | 2e–2 |
| Hellinger distance $\tau$ | 5e–4 | 2e–2 |
| Hellinger distance $\alpha$ | 9e–3 | 1e–2 |
| Time (2D) | 3 min 57 sec | 29 min 45 sec |
| Time (3D) | 4 min 13 sec | 43 min 31 sec |

Generating the ground truth distributions on $(\beta, \tau)$ via a grid search took 60 minutes. Also shown are the CPU run times for the neural and MALA schemes, both for estimating the marginals in the two-dimensional $(\beta, \tau)$ and three-dimensional case $(\beta, \tau, \alpha)$.

Using the resulting posteriors $p$, we can now generate a predicted time series with uncertainty quantification by randomly selecting $n$ samples $\hat{\mathbf{\Lambda}}^i$, $i = 1, \ldots, n$ gathered during the training process, running the noiseless ODE model Eq (6) with each sample (i.e. using a noise strength of $\sigma = 0$), and calculating the mean densities

$$\langle \hat{\mathbf{T}} \rangle = \sum_{i=1}^{N} p(\hat{\mathbf{\Lambda}}^i) \hat{\mathbf{T}}(\hat{\mathbf{\Lambda}}^i), \tag{10}$$

and analogously a standard deviation. The predicted densities are shown in Fig 1d: we see the predicted parameters capture the observed dynamics well, with all data points lying within a single standard deviation from the mean.

## Modelling the spread of COVID-19 in Berlin

We now turn to a sophisticated model of the spread of COVID-19 in Berlin, previously studied in [13]. The authors presented an extended version of the compartmental SEIRD model, modelling—among others—those infected, symptomatic, sick (i.e. requiring medical attention or hospitalisation), and critically sick (requiring ICU or otherwise urgent treatment), as well as a contact-tracing mechanism responsible for notifying those previously in contact with an infected person, and consigning them to quarantine. A model overview is presented in Fig 2, and the ODE system is similar in structure to the SIR model previously studied; see the S1 Appendix or [13] for the equations. This model was calibrated to data from an agent-based model of Berlin [31], comprising over 3 million agents, the transport system, the geography and urban structure, as well as workflow routines and travel patterns. The compartments obtained from the ABM data are exclusive, meaning that e.g. a critical agent is not also classified as symptomatic or hospitalised. The ABM was calibrated to match the case numbers of COVID-19 from February 16 to October 27 2020, and provides estimates of the hidden infection cases which were not officially recorded but nevertheless driving hospitalisation and mortality rates. Crucially, it assumes that the official infection figures recorded by the Robert-Koch Institute are the sum of the SY, H, and C compartments, and do *not* contain the I compartment, since in the early stages of the pandemic asymptomatic cases were not usually detected (cf. Fig 3).

Public health measures naturally had an impact on the virus dynamics: on March 12, factories, theatres, and concert halls started closing, and the German Bundesliga suspended all football games. Ten days later, the federal government prohibited all gatherings of more than two people, exempting single households. These measures lasted through April 2020, after which retail, schools, and kindergartens gradually started reopening, the federal government giving states broad autonomy to set their own policies on May 6. [33, 34]. Starting in mid-June,

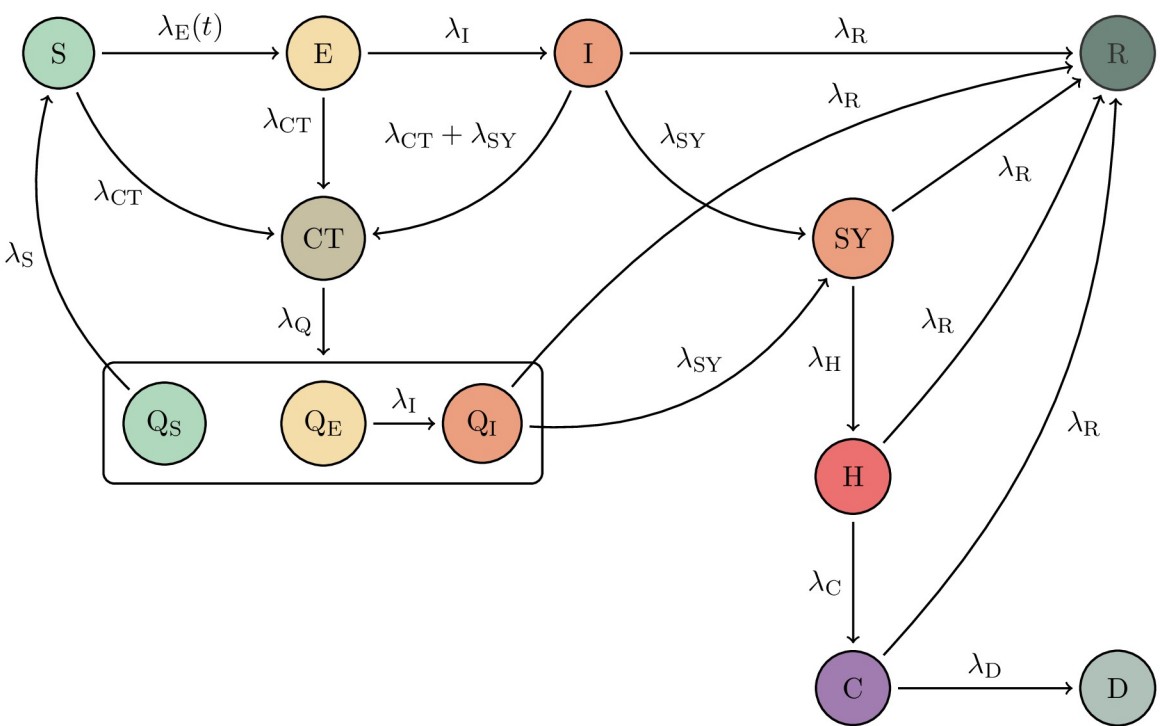

**Fig 2. Schematic illustration of the SEIRD+ model, as originally presented in [13].** Each parameter $\lambda_i$ indicates the transition rate between the respective compartments. S, E, and I are the susceptible, exposed, and infected agents. Upon contact with an infected agent, each may be contacted by the contact tracing agency (CT) and ordered to quarantine ($Q_S$, $Q_E$, $Q_I$ compartments). $\lambda_Q$ models the rate of compliance with the contact tracing agency's instructions. SY, H, and C are the symptomatic, sick, and critically sick agents. Agents from these as well as the I and $Q_I$ compartments can recover and transition to the R compartment, where they are assumed to stay, at least for the period under consideration ($< 9$ months). Finally, critically ill patients may die of the disease (D), though this compartment is not included in the loss function. We assume the exposure rate $\lambda_E$ varies as public health measures change. The parameter $\lambda_Q$ is further assumed to be a function of $\lambda_{CT}$ and CT, and is thus not learned; see S1 Appendix. Figure adapted from [13].

restrictions on social gatherings were further relaxed across the country. These measures will primarily have affected the population's exposure to the virus, and we thus assume that the exposure parameter $\lambda_E$ is piecewise linear on the intervals [Feb 16, Mar 12, Mar 22, May 6, June 15, Oct 27]. This is not taking into account virus mutations changing its infectivity or lethality. The vector of parameters $\mathbf{\Lambda}$ we wish to estimate is thus 13-dimensional, comprising the 9 parameters shown in Fig 2, one of which is time-dependent.

We split the data into a training period of 200 days, spanning the period up to September 3 ('calibration period'), and a test period, spanning the remaining eight weeks until October 27 ('projection period'). We employ a deep neural network with 3 layers, 20 neurons per layer, and the sigmoid as an activation function on all but the last layer, where we again use the modulus. The batch size $B$ is again equal to the length of the time series ($B = L$). The number of agents in each compartment span several orders of magnitude, from millions of susceptible agents down to hundreds of hospitalised or critical agents. Using the simple loss function from the previous example, Eq (7), would result in only the largest compartments being fitted accurately, since their residuals dominate the loss. We therefore scale each compartment's contribution to the total loss,

$$J = \sum_i \alpha_i \|\hat{\mathbf{T}}_i - \mathbf{T}_i\|_2^2, \tag{11}$$

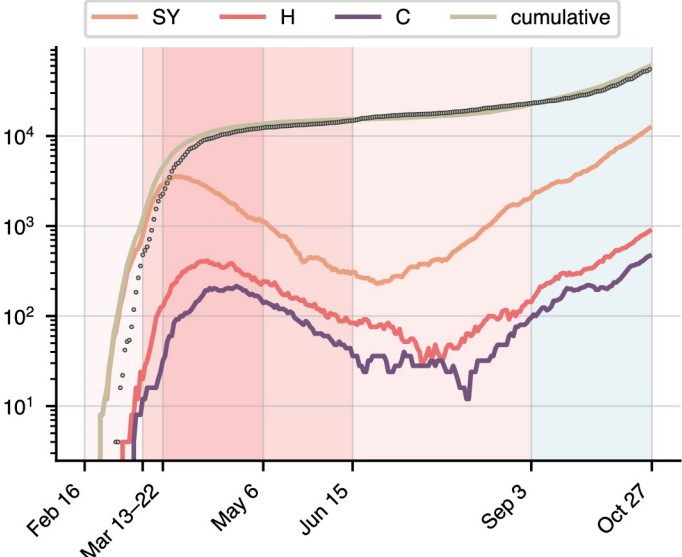

**Fig 3. Evolution of COVID-19 in Berlin for the period from February 16 to October 27 2020.** Shown are the ABM data [31] for the symptomatic, hospitalised, and critical compartments (orange, red, purple), the sum of all three (light brown, solid line), as well as the official infection figures (light brown, dots) [32]. The red period is the calibration period, with the shades representing varying levels of government restrictions and correspondingly different exposure levels $\lambda_E$: from mid-March, businesses and factories started closing; in late March, the German government imposed broad contact restrictions; in early May, schools and kindergartens started reopening across the country, followed by further loosening of restrictions in mid-June, before the start of the summer holidays. The blue period is the projection data on which we evaluate the prediction. The ABM data only contains a single Q compartment and no CT compartment. It also does not produce a D compartment, for the reasons given in the text. See S1 Appendix for details.

by choosing coefficients $\alpha_i$ that ensure all summands are roughly of equal magnitude:

$$\alpha_i^{-1} = \int_0^L \mathbf{T}_i(t)\mathrm{d}t, \tag{12}$$

where $L$ denotes the length of the time series. Note that, due to inconsistencies in the official mortality statistics for the early period of the pandemic (arising from the difficulty of discerning 'death by COVID' from 'death with COVID') we do not fit the D compartment.

Assuming uniform priors on all parameters $\lambda_i$, we show the marginal posteriors alongside the MCMC estimates in Fig 4. The means and modes of the distributions are indicated in the plot. In general, the neural posteriors are more sharply peaked and unimodal than the MCMC estimates, though the expectation values and modes tend to roughly match. A low sensitivity to $\lambda_S$ is unsurprising given the large pool of susceptible agents during the early stages of the pandemic in a city of 3.6 million. One notable exception is $\lambda_{SY}$, where the neural network predicts a much lower rate than MCMC. The marginals on $\lambda_S$ and $\lambda_{CT}$ are fairly broad, indicating low sensitivity to these transition rates. Also observe that the neural marginals for the exposure parameter are unimodal, with the means and modes obeying $\hat{\lambda}_{E,0} \gg \hat{\lambda}_{E,1} > \hat{\lambda}_{E,4} > \hat{\lambda}_{E,3} > \hat{\lambda}_{E,2}$. This is consistent with the level of government restrictions imposed, and it is interesting to note that the measures taken between March 12 and March 22 already reduced the exposure rate by two-thirds. This pattern does not hold for the MCMC estimates.

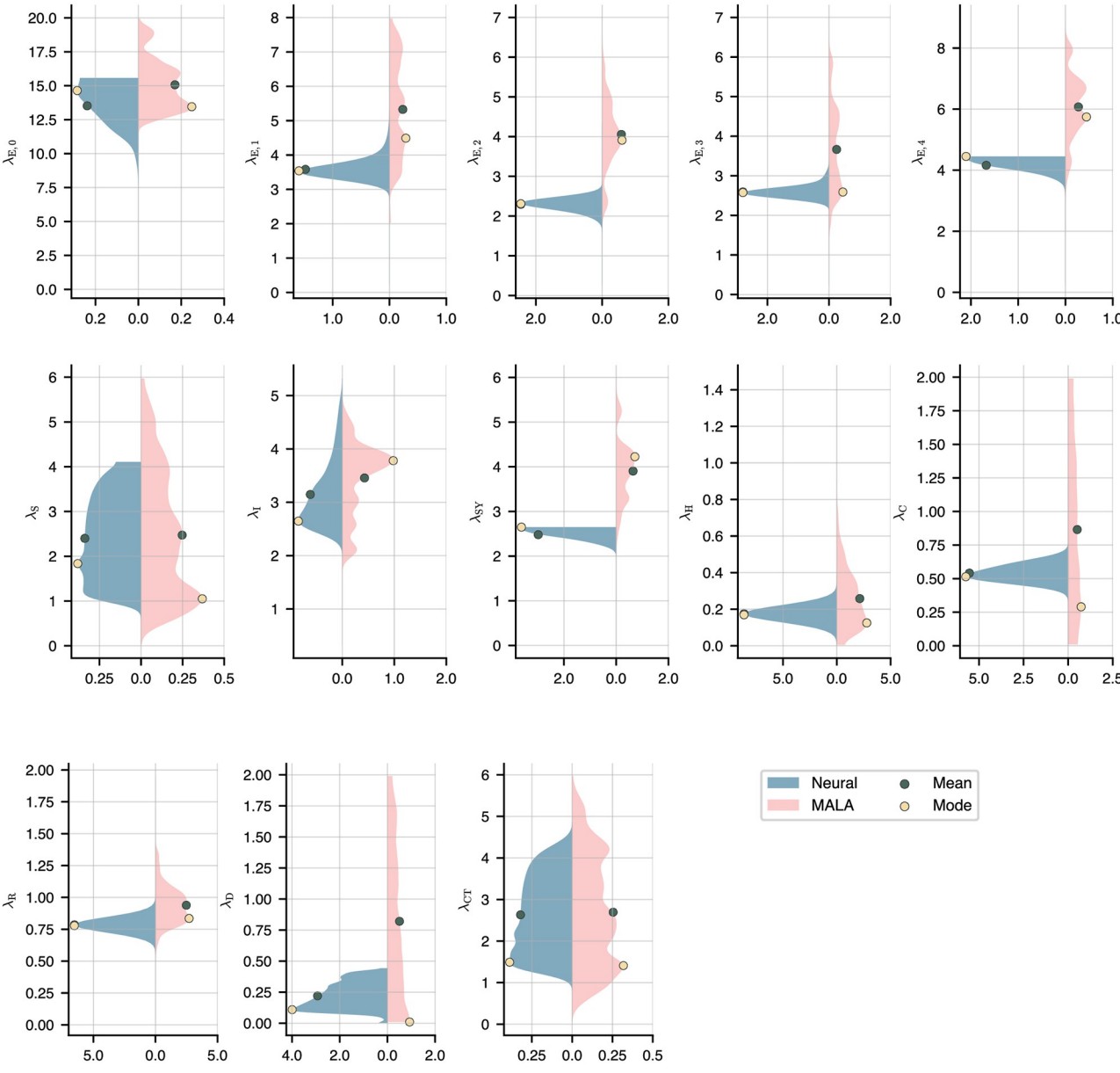

**Fig 4. Marginals on the parameters $\Lambda = (\lambda_S, \ldots, \lambda_{CT})$.** Shown are the neural marginals (blue, left side) and MCMC estimates (pink, right side), which in both cases were smoothed using a Gaussian kernel. Also shown are the means (green dots) and modes (yellow dots) of the marginals. We employ a three-layer neural network with 20 neurons per layer and sigmoid activation functions on all but the last layer, where we again use the absolute value function.

As before, we now draw $n$ samples from the joint densities to produce a mean time series $\langle \hat{\mathbf{T}} \rangle$ for each compartment. We compare the quality of the fit on each compartment, both on the training period and the projection period, using the $L^2$ residual

$$r_i^2 = \langle \hat{\mathbf{T}}_i(t) - \mathbf{T}_i(t) \rangle_t^2, \;\; i \in \{S, \ldots, Q\}, \tag{13}$$

with the expectation value $\langle \cdot \rangle_t$ taken over time. In order to circumvent having to calculate the full 13-dimensional joint distribution $p(\hat{\Lambda})$, we simply select $n = 1000$ random samples

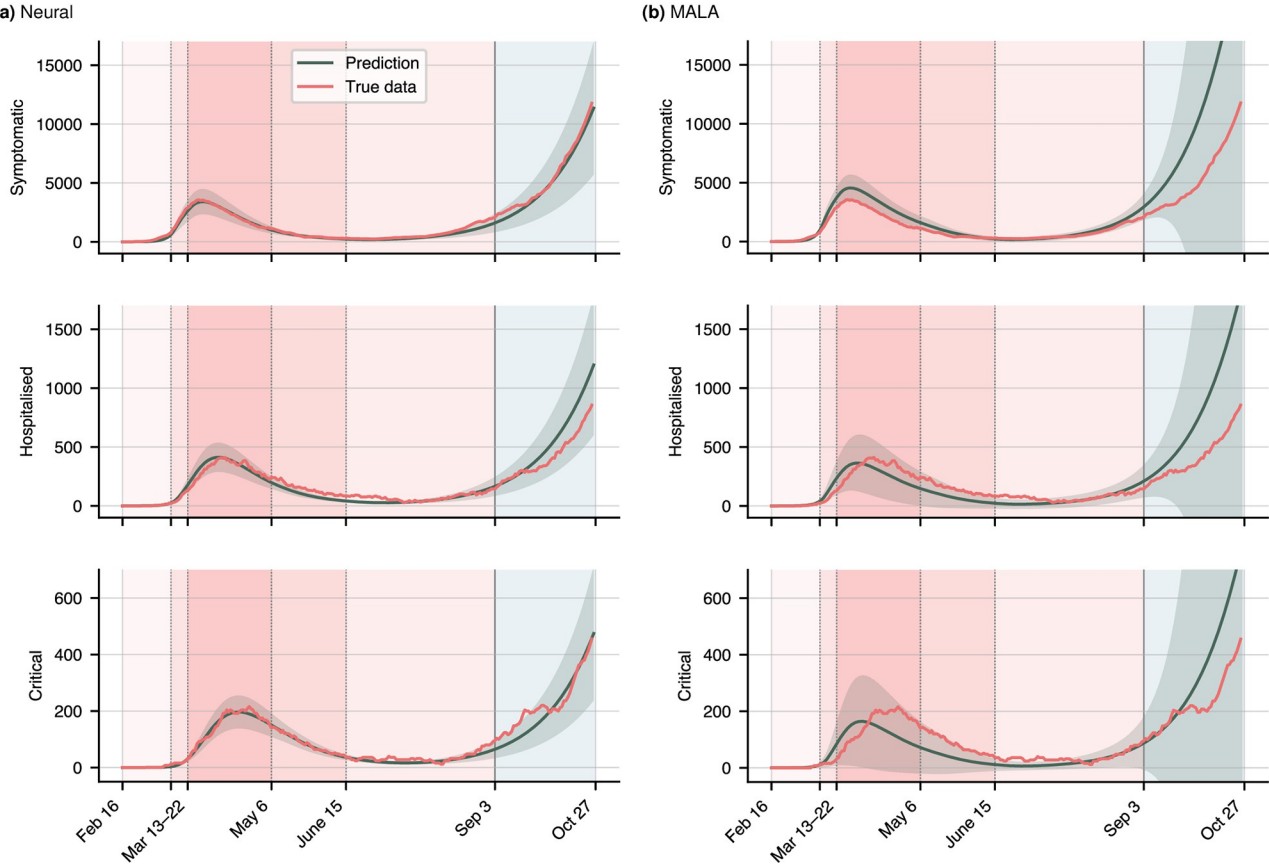

**Fig 5. Comparison of the neural calibration results (left) and the MCMC calibration results (right) for the symptomatic, hospitalized, and critical compartments.** Red lines are the true data, green lines the prediction using the estimated mean of the joint density, calculated by drawing 1000 samples from the joint distribution. The green shaded areas represent one standard deviation. The blue shaded area is the test period for which projections are generated. Calibration results for the remaining compartments are shown in S3 Fig in the S1 Appendix.

previously collected during training. Shown in Fig 5 is the true data (red), the mean prediction $\langle \hat{\mathbf{T}} \rangle$ (green), and one standard deviation (shaded area). The neural approach visibly calibrates the training data to a higher accuracy than MCMC, consequently also achieving a better fit on the projection period (blue shaded area). The residuals $r_i$ are given in Table 2. The neural scheme achieves an average calibration error of 25%, representing a 50% improvement over MCMC, and a projection error of 18%, a fourfold improvement over MCMC. At the same time, the neural network runs an order of magnitude faster.

Until now we have assumed full knowledge of all compartments in the model, but this is only due to a sophisticated and computationally expensive ABM running in the background, laboriously calibrated to official data. Without this machinery, the available data only covers the symptomatic, hospitalised, and critical compartments [32, 35, 36]. We thus re-train the model on these compartments only, and assess the calibration quality. Results are shown in Fig 6. The neural network still calibrates each compartment with an average error of 0.33, an 18% reduction compared to the full model. However, the prediction error is 0.98, an almost five-fold decrease compared to the full model. Simultaneously, the model's confidence in its predictions decreases visibly: the full model is thus required to make accurate predictions with high confidence.

**Table 2. Calibration and projection error of the neural and MALA schemes on the different compartments.**

|  | Neural error | | MALA error | |
|---|---|---|---|---|
|  | Calibration | Projection | Calibration | Projection |
| S | 0.0004 | 0.002 | 0.003 | 0.02 |
| E | 0.36 | 0.17 | 0.32 | 0.67 |
| I | 0.23 | 0.11 | 0.31 | 0.67 |
| R | 0.22 | 0.04 | 0.49 | 0.67 |
| SY | 0.25 | 0.12 | 0.35 | 0.97 |
| H | 0.32 | 0.28 | 0.52 | 0.85 |
| C | 0.28 | 0.24 | 0.64 | 0.52 |
| Q | 0.34 | 0.11 | 0.43 | 1.33 |
| **Avg.** | **0.25** | **0.18** | **0.38** | **0.71** |
| **Time** | 8 mins 45 secs | | 88 mins 1 sec | |

The calibration period refers to the test period from Feburary 16 to September 3, while the projection period is the remaining two months until October 27. The error is given in terms of the $L^2$ residual $r_i$ (Eq (13)). Also shown: CPU run time.

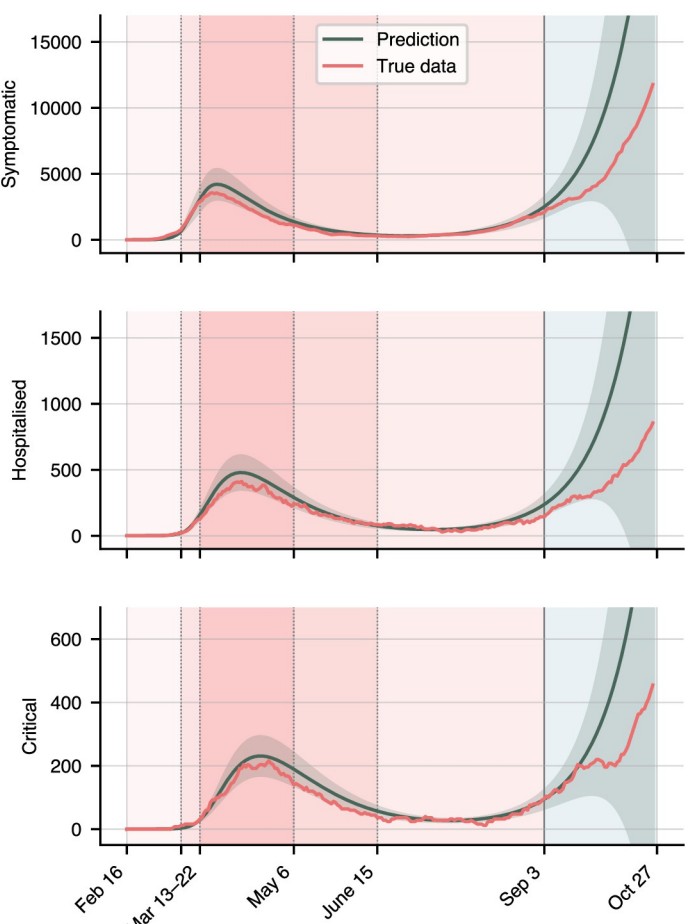

**Fig 6. Results on a reduced training dataset, consisting only of the symptomatic, hospitalized, and critical compartments.**

## Discussion

In this article we presented a method to optimally fit compartmental infection models to observations of infection spreading. We applied a novel and powerful computational method to the problem of learning probability densities on contagion parameters and providing uncertainty quantification for epidemic projections. This new methodology can be utilized based on simulation data coming from finer scale models (as demonstrated herein) or directly on observational data. In the former case, it allows finding optimal surrogate models to fine-scale infection models in order to use them in optimal control or multi-objective optimization approaches as in [13].

The strategy proposed in this paper represents a notable improvement over conventional MCMC or Langevin sampling methods due to its superior computational accuracy and efficiency in estimating the parameters of the model and their uncertainty. The comprehensive understanding of uncertainty it provides is vital to developing effective policy responses when faced with intrinsic unpredictability. We mention, in particular, that the exploration of a relatively high-dimensional parameter space using MCMC can be extremely expensive, especially when—as is the case here—the likelihood must be obtained via simulation. Furthermore, the marginals Fig 4 strongly indicate that the parameter space is highly non-convex with many different local minima trapping the MCMC sampler and significantly increasing the mixing times. In our analysis, we also noted the slow convergence of the Gelman-Rubin statistic for the Langevin sampler—see S4 Fig in the S1 Appendix. Overall, in our experiments our method delivered a 10-fold decrease in compute times, while calibrating and predicting the spread of COVID-19 significantly more accurately. In our numerical experiments, even state-of-the-art MCMC schemes fail to fully explore the parameter space, in particular if the model contains redundant parameters. Our proposed method, by contrast, does not suffer from this drawback.

Recently, new alternative sampling methods for Bayesian uncertainty quantification and inversion have been proposed; one example is the Affine Invariant Langevin Dynamics (ALDI) [37, 38], a modification of the Ensemble Kalman Sampler [39] with significant theoretical advantages over preconditioned MALA (such as affine invariance and convergence in total variation to the posterior, at least for convex problems). Further schemes include Hamiltonian Monte Carlo [20] and the bouncy particle sampler [40]. A comprehensive comparison of the various sampling schemes and their relative benefits for calibrating epidemiological models will be the subject of future work. Lastly, one current deficit of the neural parameter calibration scheme proposed in [14] is that, so far, it lacks a rigorous convergence analysis, and its theoretical properties remain unclear. This will be the subject of future work by the authors.

## Data, materials, and software availability

Code data can be found under https://github.com/ThGaskin/NeuralABM. It is easily adaptable to new models and ideas. The code uses the `utopya` package (https://utopia-project.org) [41, 42] to handle simulation configuration and efficiently read, write, analyse, and evaluate data. This means that the model can be run by modifying simple and intuitive configuration files, without touching code. Multiple training runs and parameter sweeps are automatically parallelised. The neural core is implemented using `pytorch` (https://pytorch.org).

## Supporting information

**S1 Appendix.**
(PDF)

## Acknowledgments

The authors would like to thank Hanna Wulkow for her assistance in acquiring the MODUS-Covid ABM data.

Thomas Gaskin, Tim Conrad, Grigorios A. Pavliotis and Christof Schöte designed the research and wrote the paper. Thomas Gaskin also performed the numerical experiments and wrote the code.

## Author Contributions

**Conceptualization:** Thomas Gaskin, Tim Conrad, Grigorios A. Pavliotis, Christof Schütte.

**Methodology:** Thomas Gaskin.

**Software:** Thomas Gaskin.

**Writing – original draft:** Thomas Gaskin, Tim Conrad, Grigorios A. Pavliotis, Christof Schütte.

**Writing – review & editing:** Thomas Gaskin.

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
