## [Decision Letter · Decision Letter 0]

3 Jun 2024

PONE-D-23-41302Neural parameter calibration and uncertainty quantification for epidemic forecastingPLOS ONE

Dear Dr. Gaskin,

Thank you for submitting your manuscript to PLOS ONE. After careful consideration, we feel that it has merit but does not fully meet PLOS ONE’s publication criteria as it currently stands. Therefore, we invite you to submit a revised version of the manuscript that addresses the points raised during the review process.

We look forward to receiving your revised manuscript.

Kind regards,

Viswanathan Arunachalam, Ph.D.

Academic Editor

PLOS ONE

Journal Requirements:

 [TG was funded by the University of Cambridge School of Physical Sciences VC Award via DAMTP and the Department of Engineering, and supported by EPSRC grants EP/P020720/2 and EP/R018413/2. TC and CS were funded by the Deutsche Forschungsgemeinschaft (DFG) under Germany's Excellence Strategy through grant EXC-2046  \\emph{The Berlin Mathematics Research Center} MATH+ (pro\\-ject no. 390685689) and via the grant MODUS-COVID by the German Federal Ministry for Education and Research. GP is partially supported by the Frontier Research Advanced Investigator Grant ERC grant Machine-aided general framework for fluctuating dynamic density functional theory. ].  

[TG was funded by the University of Cambridge School of Physical Sciences VC Award

via DAMTP and the Department of Engineering, and supported by EPSRC grants

EP/P020720/2 and EP/R018413/2. TC and CS were funded by the Deutsche

Forschungsgemeinschaft (DFG) under Germany’s Excellence Strategy through grant

EXC-2046 The Berlin Mathematics Research Center MATH+ (project no. 390685689)

and via the grant MODUS-COVID by the German Federal Ministry for Education and

Research. GP is partially supported by the Frontier Research Advanced Investigator

Grant ERC grant Machine-aided general framework for fluctuating dynamic density

functional theory. The authors would also like to thank Hanna Wulkow for her support

in acquiring the ABM data.]

 [TG was funded by the University of Cambridge School of Physical Sciences VC Award via DAMTP and the Department of Engineering, and supported by EPSRC grants EP/P020720/2 and EP/R018413/2. TC and CS were funded by the Deutsche Forschungsgemeinschaft (DFG) under Germany's Excellence Strategy through grant EXC-2046  \\emph{The Berlin Mathematics Research Center} MATH+ (pro\\-ject no. 390685689) and via the grant MODUS-COVID by the German Federal Ministry for Education and Research. GP is partially supported by the Frontier Research Advanced Investigator Grant ERC grant Machine-aided general framework for fluctuating dynamic density functional theory. ]

6. We notice that your supplementary figures are uploaded with the file type 'Figure'. Please amend the file type to 'Supporting Information'. Please ensure that each Supporting Information file has a legend listed in the manuscript after the references list.

Additional Editor Comments:

Dear Dr. Thomas Gaskin

Thank you for submitting your manuscript to PLOS One  

I have completed my evaluation of your manuscript. The reviewers recommend reconsideration of your manuscript following minor revision and modification. I invite you to resubmit your manuscript after addressing the comments below. Please resubmit your revised manuscript by August 1, 2024.

Reviewer 1:

The paper is interesting and well written. The methodological proposal is not new, and I ask the authors to clarify this point in the paper. The novelty is that they apply such an approach to estimate the parameters of ordinary differential equations.

In the revision, I address some issues to add details to the manuscript. I also checked the link to the code which is available on Github and it seems well written, ensuring replicability of some results. Note, the methodological part is quite technical and I am not sure if the general readers of the journal would be interested in all the details. However, from a statistician's perspective, these technicalities are valuable. I suggest considering placing them in the Appendix.

I recommend a minor revision.

Reviewer 2 :

This paper uses neural network to directly solve the inverse problem of getting epidemic parameters in compartment models for infectious diseases and quantify the associated uncertainty. Compared with traditional MCMC approaches, the neural scheme, proposed and applied in [13] and [14], is more efficient and effective. This is demonstrated in a simulation and more importantly in a study of COVID-19 in Berlin in 2020. The paper is clearly written and well supported with numerical evidences. I would recommend acceptance given some minor issues can be addressed.

1. Even though it is an application of the method proposed elsewhere [13][14], it would be good to highlight why it yields a valid posterior distribution, especially 'in the limit of infinitely many chains'?

2. When comparing running times, e.g. in Table 1, does time of the nueral scheme include the neural network training time?

3. On line 222, it should be 'estimate' instead of 'estimates'.

Reviewers' comments:

Reviewer's Responses to Questions

**Comments to the Author**

1. Is the manuscript technically sound, and do the data support the conclusions?

Reviewer #1: Yes

Reviewer #2: Yes

2. Has the statistical analysis been performed appropriately and rigorously? 

Reviewer #1: Yes

Reviewer #2: Yes

3. Have the authors made all data underlying the findings in their manuscript fully available?

Reviewer #1: No

Reviewer #2: Yes

4. Is the manuscript presented in an intelligible fashion and written in standard English?

Reviewer #1: Yes

Reviewer #2: Yes

5. Review Comments to the Author

Reviewer #1: The authors introduce a machine learning strategy rooted in the Bayesian framework that has been previously applied in other contexts. They consider ordinary differential equations and propose a neural network approach to estimate model parameters bypassing the requirement for Riemannian metrics. The loss function incorporates knowledge of model equations, and likelihoods are estimated via simulation. By conducting multiple parallelizable training iterations, the methodology focuses on the estimated parameters' accuracy. 

Minor issues:

- Introduction: When the authors mention the importance of uncertainty, new ensemble methods have been proposed, and they seem to improve accuracy in such a context. See Sherratt et al. (2023), among others. A mention of such approaches that are proposed to ensemble predictions of various models should be made in the introduction or the discussion. 

Sherratt, K., et al. (2023). Predictive performance of multi-model ensemble forecasts of COVID-19 across European nations, eLife. https://elifesciences.org/articles/81916.

- Page 2: The authors state that the method â€œhas previously been applied to a diverse set of problems.â€ specify where and which kind of problems, providing proper citations. It is important to clearly state here that the proposed neural network is not new, which is the novelty here. 

- Page 8, write figures 2 and 4 with capital F. 

- On page 9, write Table 2 with capitals t. 

- Appendix Page 12: In the caption of Figure S3, write Table 2 with capital t.

Reviewer #2: This paper uses neural network to directly solve the inverse problem of getting epidemic parameters in compartment models for infectious diseases and quantify the associated uncertainty. Compared with traditional MCMC approaches, the neural scheme, proposed and applied in [13] and [14], is more efficient and effective. This is demonstrated in a simulation and more importantly in a study of COVID-19 in Berlin in 2020. The paper is clearly written and well supported with numerical evidences. I would recommend acceptance given some minor issues can be addressed.

1. Even though it is an application of the method proposed elsewhere [13][14], it would be good to highlight why it yields a valid posterior distribution, especially 'in the limit of infinitely many chains'?

2. When comparing running times, e.g. in Table 1, does time of the nueral scheme include the neural network training time?

3. On line 222, it should be 'estimate' instead of 'estimates'.

6. PLOS authors have the option to publish the peer review history of their article (what does this mean?). If published, this will include your full peer review and any attached files.

Reviewer #1: No

Reviewer #2: **Yes: **Shiwei Lan

---

## [Author Response · Author response to Decision Letter 0]

20 Jun 2024

One reviewer ticked 'no' under Question 3: Have the authors made all data underlying the findings in their manuscript fully available? We are unsure why this is: all data and code are fully and freely available at the GitHub, and is versioned with a permanent DOI. The repository comes with a detailed README, detailing exactly how to reproduce the plots and run the numerical experiments shown in this article.

---

## [Editor Report · Decision Letter 1]

24 Jun 2024

Neural parameter calibration and uncertainty quantification for epidemic forecasting

PONE-D-23-41302R1

Dear Dr. Gaskin,

We’re pleased to inform you that your manuscript has been judged scientifically suitable for publication and will be formally accepted for publication once it meets all outstanding technical requirements.

Kind regards,

Viswanathan Arunachalam, Ph.D.

Academic Editor

PLOS ONE
---

## [Editor Report · Acceptance letter]

28 Jun 2024

PONE-D-23-41302R1 

PLOS ONE

Dear Dr. Gaskin, 

I'm pleased to inform you that your manuscript has been deemed suitable for publication in PLOS ONE. Congratulations! Your manuscript is now being handed over to our production team.

Kind regards, 

on behalf of

Dr. Viswanathan Arunachalam 

Academic Editor

PLOS ONE